# Heart Failure Differentially Modulates Natural (Sinoatrial Node) and Ectopic (Pulmonary Veins) Pacemakers: Mechanism and Therapeutic Implication for Atrial Fibrillation

**DOI:** 10.3390/ijms20133224

**Published:** 2019-06-30

**Authors:** Chao-Shun Chan, Yung-Kuo Lin, Yao-Chang Chen, Yen-Yu Lu, Shih-Ann Chen, Yi-Jen Chen

**Affiliations:** 1Graduate Institute of Clinical Medicine, College of Medicine, Taipei Medical University, Taipei 11042, Taiwan; 2Division of Cardiology, Department of Internal Medicine, Taipei Medical University Hospital, Taipei 11042, Taiwan; 3Division of Cardiology, Department of Internal Medicine, School of Medicine, College of Medicine, Taipei Medical University, Taipei 11042, Taiwan; 4Division of Cardiovascular Medicine, Department of Internal Medicine, Wan-Fang Hospital, Taipei Medical University, Taipei 11696, Taiwan; 5Department of Biomedical Engineering, National Defense Medical Center, Taipei 11490, Taiwan; 6Division of Cardiology, Department of Internal Medicine, Sijhih Cathay General Hospital, New Taipei City 22174, Taiwan; 7School of Medicine, College of Medicine, Fu-Jen Catholic University, New Taipei City 24257, Taiwan; 8Division of Cardiology, Department of Medicine, Taipei Veterans General Hospital, Taipei 11217, Taiwan; 9Institute of Clinical Medicine, and Cardiovascular Research Center, National Yang-Ming University, Taipei 11221, Taiwan; 10Cardiovascular Research Center, Wan-Fang Hospital, Taipei Medical University, Taipei 11696, Taiwan

**Keywords:** heart failure, atrial fibrillation, sinoatrial node dysfunction, pulmonary veins, sinoatrial node

## Abstract

Heart failure (HF) frequently coexists with atrial fibrillation (AF) and dysfunction of the sinoatrial node (SAN), the natural pacemaker. HF is associated with chronic adrenergic stimulation, neurohormonal activation, abnormal intracellular calcium handling, elevated cardiac filling pressure and atrial stretch, and fibrosis. Pulmonary veins (PVs), which are the points of onset of ectopic electrical activity, are the most crucial AF triggers. A crosstalk between the SAN and PVs determines PV arrhythmogenesis. HF has different effects on SAN and PV electrophysiological characteristics, which critically modulate the development of AF and sick sinus syndrome. This review provides updates to improve our current understanding of the effects of HF in the electrical activity of the SAN and PVs as well as therapeutic implications for AF.

## 1. Introduction

Heart failure (HF) and atrial fibrillation (AF) are increasing endemic and frequently coexist in part due to common risk factors, such as age, diabetes, hypertension, coronary artery disease and valvular heart disease [1,2], and predispose to each other [3]. According to the results of the Framingham Heart Study, 26% of patients with AF had a concurrent HF diagnosis. Similarly, 24% of patients with HF had a concurrent diagnosis of AF [4]. AF precipitates HF through the loss of atrial contraction, rapid and irregular ventricular rhythm, and decrease in coronary flow reserve [5]. By contrast, HF precipitates AF by contributing to atrial electrical and structural remodeling due to adrenergic stimulation, neurohormonal activation, abnormal intracellular calcium handling, and atrial stretch [6,7]. AF has been identified as a crucial predictor of mortality in patients with HF [8,9,10]. This is particularly relevant to patients with less advanced HF and recent AF onset [11]. A recent registry study showed that AF increased the risk for the composite of mortality and HF hospitalization in HF patients with preserved or mid-range ejection fraction, but not in HF patients with reduced ejection fraction after multivariable adjustment [12]. However, HF classification simply by ejection fraction may not be optimal [13]. Determination of ejection fraction from echocardiography is unreliable with intra and interobserver variability, and without consideration of loading condition. There were several potential HF classification methods with considerations for identifying pathophysiological mechanisms and underlying etiologies [14]. Therefore, using alternate HF classification schemes in assessing the prognostic impact of AF on HF may lead to different conclusions. An AF treated with rate control is not associated with more unfavorable clinical outcomes compared to a patient with sinus rhythm maintained and protected with the use of the rhythm control strategy [15]. It is possible to adopt strategies that limit the damage caused by AF such as standard HF therapy and stroke prevention [16]. The presence of HF significantly affects therapeutic considerations for AF regarding treatment strategy and medications for rate or rhythm control as well as the outcomes of cardioversion and catheter ablation [7].

Pulmonary veins (PVs) contain cardiomyocytes and are ectopic pacemakers. PVs represent the most crucial AF inducers [17,18,19]; PVs act as triggers on a susceptible substrate or fire rapidly as AF-maintaining drivers [20]. AF is associated with electrical and structural remodeling in PVs and atrial substrates [21,22]. Moreover, dysfunction of the sinoatrial node (SAN), the natural pacemaker, plays a critical role in the pathophysiology of AF [23,24]. SAN dysfunction enhances PV arrhythmogenesis, which may increase the risk of AF [25,26]. HF affects the SAN and PVs differently. By modulating the electrical activity of PVs and the SAN, HF may induce AF and SAN dysfunction. The present study is a review of the relevant literature for updating our current understanding of the crosstalk between PVs and the SAN, the role of HF in the distinctive electrical properties of PVs and the SAN, and therapeutic implications for AF.

## 2. Distinct Electrophysiological and Structural Characteristics of PVs

PVs contain myocardial sleeves extending from the left atrium (LA) [27,28]. The length of myocardial sleeves varies considerably from species to species, but it may be 1–4 cm in the human heart [29]. Previous histological examination in autopsy specimens has shown that PV myocardial sleeves were found in 100% of patients with AF, compared to 85% of patients without AF [30]. The recent studies from voltage map showed that the lengths of myocardial sleeves are longer in the left and right superior PVs but markedly shorter in the left and right inferior PVs [31,32]. Patients with AF have significant longer myocardial sleeves [30]. The length of PV myocardial sleeves correlates positively with male sex, body mass index, and body surface area [31,32]. PVs contain cardiomyocytes with arrhythmogenic activity due to enhanced automaticity, genesis of triggered activity, and induction of micro-re-entry, which are crucial for the initiation and maintenance of AF [18,33]. Alterations in the expression levels and functions of cardiac ion channels, abnormal calcium handling, and unique structural characteristics play critical roles in PV arrhythmogenesis [34].

PV cardiomyocytes have distinct electrophysiological characteristics from those of LA cardiomyocytes. PV pacemaker cardiomyocytes exhibit a funny pacemaker current (I*_f_*) [33]. The fast PV pacemaker cardiomyocytes have a larger I*_f_* than the slow PV pacemaker cardiomyocytes [35]. PV cardiomyocytes have a considerably lower inward rectifier potassium current (I_K1_) than do LA cardiomyocytes, which facilitates pacemaker depolarization due to more positive resting membrane potential [36]. Moreover, the T-type calcium current (I_Ca-T_) causes the release of calcium from the sarcoplasmic reticulum (SR) at a low voltage, thus enabling the generation of pacemaker activity. I_Ca-T_ has been demonstrated to be relatively large in the pacemaker cardiomyocytes of PVs than in those of the LA [37]. The electrophysiological properties of cardiomyocytes in PVs are characteristic of enhanced automaticity.

PV pacemaker cardiomyocytes have lower I_K1_. A low I_K1_ reduces the resting membrane potential, which inactivates sodium channels and causes slow conduction, together with abrupt changes in fiber orientation that promote unidirectional block, slow conduction, and facilitate re-entry. Previous studies have shown that PV cardiomyocytes may exhibit lower L-type calcium current (I_Ca-L_) than neighboring LA cardiomyocytes, which leads to a reduction in the action potential duration (APD) and refractory period [38]. Moreover, rapid atrial pacing was reported to induce fast PV spontaneous activity, a short APD, large I*_f_* and I_ti_, and a high incidence of early afterdepolarization (EAD) and delayed afterdepolarization (DAD) [39]. 

Connexins (Cxs) are responsible for electric coupling between cardiomyocytes [40]. Reduced synthesis of Cxs, which are gap junction proteins, was demonstrated to contribute to arrhythmia development [32]. PV cardiomyocytes have a lower density of Cx40 than adjacent LA cardiomyocytes, implying that impaired electrical coupling may result in slow conduction and promote re-entry [41].

### 2.1. Autonomic Nervous System in PV Electrical Activity

PVs receive extensive autonomic innervation [20]. Cardiac autonomic inputs pass across the epicardial ganglionated plexuses, which are located close to the PV ostia. Both sympathetic and parasympathetic nerves exist in the same location and exhibit intrinsic activities, which are independent of extrinsic neural inputs [42]. The stimulation of the autonomic nervous system induces PV arrhythmogenesis. Isoproterenol accentuates spontaneous activity in PVs, and by contrast, acetylcholine hyperpolarizes the membrane and attenuates spontaneous activity [39]. Moreover, isoproterenol was shown to induce EAD and DAD in PVs [43]. Stress disorder, such as anxiety is an important risk factor of AF [44]. Patients with stress disorder may have increased activity of sympathetic nervous system, inducing PV arrhythmogenesis and promoting the onset, progression, and maintenance of AF.

### 2.2. Calcium Homeostasis in PV Cardiomyocytes

Abnormal calcium handling plays a crucial role in PV arrhythmogenesis [34]. Compared with those without isoproterenol-induced EAD, PV cardiomyocytes with isoproterenol-induced EAD exhibit a larger increase in the I_Ca-L_ after isoproterenol stimulation [43]. I_Ca-T_ is larger in PV pacemaker cardiomyocytes than in PV non-pacemaker cardiomyocytes or LA cardiomyocytes [37]. An increase in the transient inward current (I_ti_) and sodium/calcium exchange (NCX) current was shown to enhance EAD in canine PVs [45]. PV electrical activity was reported to be reduced by KB-R7943 (an NCX inhibitor), which reduces the I_ti_ amplitude and SR calcium store [46]. Calcium influx from inward NCX, I_Ca-L_ and I_Ca-T_ can trigger a release of large amounts of calcium from the SR; these findings indicate that abnormal calcium handling plays a crucial role in PV arrhythmogenesis. Moreover, dysfunction of the ryanodine receptor (RyR) causes a diastolic calcium leak and activates a calcium spark, which lead to membrane depolarization and DADs. Studies have demonstrated that a low dose of ryanodine can induce PV burst firings [47]; FK-506, which dissociates the RyR-FKBP 12.6 complex and inhibits calcineurin activity, can induce RyR dysfunction and PV burst firings [19]. By contrast, K201 (an RyR stabilizer) may reduce the diastolic calcium leak, which causes a reduction in the PV burst firing rate, DADs, and I_ti_ [48]. An increase in the SR calcium store and calcium spark with the activation of NCX induces DADs and enhances PV arrhythmogenic activity [49]. Accordingly, abnormal intracellular calcium handling may play a pivotal role in PV arrhythmogenesis.

### 2.3. Role of Renin Angiotensin System in PV Electrical Activity

PVs are affected by the activation of the atrial renin angiotensin system (RAS). Angiotensin II increases I*_f_* and inhibits I_k1_, resulting in increased PV automaticity [50]. Moreover, angiotensin II increases the I_Ca-L_, delayed rectifier potassium current, I_ti_, and NCX current, thus enhancing triggered activity in PV cardiomyocytes. The increased automaticity and triggered activity in PV cardiomyocytes were reported to be attenuated by pretreating cardiomyocytes with angiotensin II receptor blockers [50]; this finding suggests the critical role of RAS in the PV arrhythmogenesis and the pathophysiology of AF. Additionally, a direct renin inhibitor (aliskiren) reduces the PV spontaneous activity and I_Ca-L_ and causes a decrease in the calcium content in PV cardiomyocytes [51]. Accordingly, activation of RAS directly affects PV arrhythmogenesis through the dysregulation of calcium homeostasis.

### 2.4. Mechanoelectrical Feedback on PV Arrhythmogenesis

The mechanoelectrical feedback indicates a phenomenon in which a mechanical load on the myocardial tissue changes the electrical activity of cardiomyocytes because of increased automaticity or triggered activity [52]. Patients with AF were found to have dilated PVs compared with those without AF [53,54,55]. PV dilatation not only provides structural support for re-entry but also significantly changes the electrical properties of PVs [56]. Dilated PVs were reported to be associated with a high stretch level, which may induce membrane depolarization and prolong APD in isolated PV cardiomyocytes [52,57]. The stretch increases the PV firing rate and increases the incidence of spontaneous and triggered activities of PVs (Figure 1) [58]. These arrhythmogenic effects caused by high stretch levels in PVs were reported to be attenuated by stretch-activated ion channel blockers, gadolinium and streptomycin. These findings indicate that stretch-induced PV arrhythmogenesis may contribute to AF development [59]. Significant dilation of both superior PVs was demonstrated in patients with AF; however, only 28% of the trigger foci arose from the largest PVs [57]. Previous studies have shown that higher number of AF foci arise from relatively longer PV myocardial sleeves [17,18], suggesting a relationship between the extent of PV myocardial sleeves and ectopic foci of AF initiation. It was speculated that the anatomic and geometric differences of PVs may participate in the firing of PVs in AF, and PV size may not sufficiently predict the origin of AF firing [57].

### 2.5. Interaction of PV Cardiomyocytes and Fibrosis

The extent of fibrosis in myocardial sleeves in PVs was greater in patients with AF than in those without AF [29]. Information regarding mechanisms underlying fibrosis development within PVs in individuals without underlying heart disease, traditional risk factors, or a long history of AF is limited. Cell loss with interstitial fibrosis replacement is frequently found in the aged LA [60]. Fibrosis of PVs induces an increase in resistance to traveling electrical impulses as well as slow conduction; together with the nonuniform anisotropy, increased resistance may cause re-entry and play a critical role in PV arrhythmogenesis [61]. Furthermore, collagen, the major element of the fibrotic tissue, could directly increase the I*_f_*, late sodium current (I_Na-Late_), I_K1_, and small-conductance calcium-activated potassium current, leading to enhanced automaticity and triggering the PV activity [62].

## 3. Distinct Electrophysiological and Structural Characteristics of the SAN

The SAN, composed of clusters of pacemaker myocytes, is a crescent-shaped structure located at the junction of the superior vena cava and right atrium (RA) along the sulcus terminalis. The SAN complex is formed by pacemaker cells interspersed with nerves and capillaries, scaffolded by dense connective tissue [63]. The interspersed fibrous matrix with surrounding fatty insulation of the SAN provides insulation and prevents the suppression of pacemaker automaticity from the surrounding atrial myocardium [64]. The position of the leading pacemaker site in the SAN shifts depending on numerous conditions, such as autonomic nerve stimulation, age, and underlying heart diseases [64]. The more superior the position of the leading pacemaker site, the higher the heart rate is. Stimulation of the sympathetic nervous system causes the leading pacemaker site to shift to a relatively superior position and results in an increase in the heart rate [65].

Numerous ion currents are involved in the activation of the SAN. Action potentials with a slow upstroke initiated in the SAN center spread peripherally into the musculature of the terminal crest. The main inward current in the center of the SAN is the I_Ca-L_, whereas the sodium current (I_Na_) operates in the periphery of SAN for providing a sufficient inward current to depolarize the atrial tissue [66]. The I*_f_* triggers spontaneous and repetitive diastolic depolarization to activate I_Ca-L_ within the SAN. The absence of I_K1_ in the SAN enables membrane repolarization below the I*_f_* threshold. The slow decay of I_Kr_ and I_Ks_ results in a slow downstroke of SAN action potentials [66].

The distribution of Cx channel varies within the SAN. In the center of the SAN, electrical coupling is weak because Cx40 and Cx43, which form large and medium conductance channels, are sparingly expressed or absent. However, Cx45, which forms small-conductance channels, is expressed in the center of the SAN. However, in the periphery of the SAN, Cx40, Cx43, and Cx45 are all present because strong electrical coupling is needed to drive the atrial myocardium [40].

### 3.1. SAN Dysfunction in AF

SAN dysfunction is common in patients with AF [67]. In a canine model with pacing-induced AF, persistent rapid atrial pacing for more than two weeks resulted in SAN dysfunction characterized by a slow intrinsic heart rate and prolonged SAN recovery time, which gradually recovered after termination of rapid atrial pacing [68]. In human volunteers, rapid atrial pacing for only 10 to 15 minutes was reported to impair SAN function, which suggests that short durations of atrial pacing or paroxysmal episodes of AF are associated with SAN remodeling and SAN dysfunction in humans [69]. Accordingly, AF can result in SAN dysfunction. Electrical, structural, and autonomic remodeling should contribute to SAN dysfunction in patients with AF. In a canine model, atrial tachypacing has been shown to downregulate the mRNA expression of HCN4 and reduce SAN I*_f_* [70], suggesting that AF results in electrical remodeling of SAN. Furthermore, AF induced by rapid atrial pacing is associated with atrial structural change characterized by marked bi-atrial dilation, an increase in mitochondrial size and number, and disruption of the SR [71]. The atrial dilation combined with rapid atrial rate may predispose atrial ischemia and SAN dysfunction [72]. In addition, loss of muscle fibers in the SAN was found in AF patients in an autopsy study [73]. The structural remodeling of SAN from repeated episodes of AF or prolonged persistence of AF can result in atrial cardiomyocyte apoptosis with progressive atrial fibrosis and dilation [66]. Accordingly, structural remodeling of SAN can contribute to SAN dysfunction in patients with AF. 

Autonomic nervous system is a major regulatory factor of SAN automaticity and sinoatrial conduction. Autonomic dysfunction is a common cause of SAN dysfunction [74]. Adrenergic or cholinergic dysregulation may contribute to pacemaker and conduction abnormality within the SAN [75]. Tachycardia-bradycardia syndrome is the extreme expression of a continuum, characterized by a substantial loss of integrity of the SAN function. AF is a disease also caused by a defective function of the SAN. If SAN could be less torpid, it would be capable of antagonizing the tendentially predominating activity of PVs, and AF would not arise.

### 3.2. Crosstalk Between PVs and the SAN

Slow heart rate predicts occurrence of AF [76,77] and recurrence of AF after catheter ablation [78]. Clinical studies have shown that SAN dysfunction is frequently associated with the genesis of AF and atrial flutter, and tachycardia-bradycardia syndrome [67,79]. Up to 50% of patients with SAN dysfunction are accompanied by AF [80,81]. It is speculated that SAN dysfunction and AF share similar risk factors and pathophysiological processes [82].

Electrical activity of the SAN has been demonstrated to modulate PV arrhythmogenesis. In a guinea-pig model, the action potentials recorded from PVs were dominated by the SAN, and rapid pacing in PVs could overdrive the SAN [83]. This laboratory evidence suggested putative crosstalk between PVs and the SAN. An animal study revealed that when the connection between the SAN and PVs was disrupted, PVs exhibited a higher number of burst firings and triggered activity in response to provocative agents than did the control SAN-PV preparation in which the SAN-PV connection was intact (Figure 2) [26]. Moreover, heptanol (a gap junction inhibitor) was demonstrated to modulate the electrical activity of the SAN and PVs, as evidenced by a reduction in the beating rate of the SAN and the induction of PV burst firings [84]. Accordingly, a decrease in the electric activity of the SAN and cellular uncoupling not only results in SAN dysfunction but also facilitates PV arrhythmogenesis by causing a loss of overdrive suppression from the SAN, which may result in the development of AF. PVs exhibit different electrophysiological properties compared with the SAN. Table 1 summarizes the distinct electrophysiological characteristics of PVs and the SAN.

### 3.3. HF Differentially Modulates Electrical Activity in the SAN and PVs

HF induces significant changes in the atrium, which facilitate the development of AF. Various mechanisms, including atrial stretch, abnormal calcium handling, autonomic and neuroendocrine dysfunction, play a critical role in the pathophysiology of AF [11]. These changes cause a reduction in the atrial refractory period, retard atrial conduction, or increased heterogeneous repolarization, thus producing a substrate for the initiation and perpetuation of AF [11]. Rhythm control with antiarrhythmic drugs is not superior to rate control in patients with concomitant HF and AF [15]. However, catheter ablation has been associated with positive outcomes in patients with coexisting AF and HF [85,86,87,88]. In the CASTLE-AF trial, all-cause mortality, cardiovascular death, and hospitalization for HF were significantly reduced by catheter ablation (PV isolation: 100%, additional lesions: 51.7%), suggesting that PV trigger might play a crucial role in the initiation of AF in patients with HF [88]. However, the detrimental role that the massive destruction of atrial cardiomyocytes by radiofrequency can play, thus disturbing the mechanical activity of atrial chamber, is the region that radiofrequency ablation would aim to protect. Catheter ablation can cause further injury to the LA and impair the reservoir, conduit, and transport functions of the LA. The benefit-to-risk of catheter ablation in patients with HF remains to be established [89].

By using a conventional microelectrode system, in our studies, we have reported that HF PVs exhibited higher beating rates in isolated PV preparations than control PV preparations, and a higher incidence of DADs was observed in HF PVs but not in control PVs [90,91]. Moreover, with a multi-electrode array system, a higher incidence of high-frequency irregular electrical activity was recorded in the HF PVs than in the HF LA; high-frequency irregular electrical activity was not observed in control PVs and LA [92]. The high-frequency irregular electrical activity of HF PVs might be associated with a higher incidence of accelerated spontaneous activity, DAD and EAD, and a higher depolarized resting membrane potential compared with control PVs [92]. These findings suggest that HF can induce arrhythmogenesis through enhanced automaticity and triggered activity, and PVs play a crucial role in the initiation and maintenance of AF in HF.

Patients with HF exhibit significant remodeling of the SAN function characterized by a decrease in the intrinsic heart rate, prolonged corrected SAN recovery time and sinoatrial conduction time, and a caudal shift of the leading pacemaker site [93]. The decreased intrinsic heart rate and prolonged corrected SAN recovery time result from decreased diastolic depolarization rate, which are attributed to attenuated I*_f_* [94]. The decrease in I*_f_* in HF is associated with mRNA and protein downregulation of HCN4 [95]. In HF, the SAN exhibits a lower expression level of sodium channel protein, which may reduce the pacemaker rate and sinoatrial conduction velocity. The effects of tetrodotoxin (an I_Na_ inhibitor) on reducing the SAN pacemaker rate and action potential amplitude are more prominent in the SAN in HF, thus suggesting that impaired I_Na_ causes SAN dysfunction in HF [96].

HF may enhance PV automaticity and triggered activity through the overactivation of sympathetic nerve system, which may partly contribute to the development of AF in HF. Our study revealed that isoproterenol increased PV firing rate and reversed the direction of the electrical conduction between the SAN and PVs [90], which is different from the known effects of isoproterenol in healthy SAN-PV tissue preparation, whereas isoproterenol increases SAN and PV beating rates without changing SAN-PV electrical conduction. Accordingly, overactivation of sympathetic nerve system may have a greater effect on PVs than the SAN in HF.

### 3.4. HF Differentially Induces Calcium Homeostasis Dysregulation in PVs and the SAN

HF PV cardiomyocytes exhibited an increase in I_ti_, the calcium store in the SR, and the spontaneous calcium leak level, and a higher diastolic calcium concentration [88]. An increase in diastolic calcium concentrations under SR calcium overload might promote NCX activity, activate I_ti_, induce DAD, and lead to PV arrhythmogenesis in HF. Moreover, as compared with control PV cardiomyocytes, HF PV cardiomyocytes have a greater width, longer duration, and longer decay time of the calcium transient (Figure 3) [91]. In addition, HF PV cardiomyocytes demonstrated a decrease in I_Na_, an increase in I_Na-Late_, and a reverse mode of NCX current [91], which may lead to calcium overload and higher arrhythmogenesis in PV cardiomyocytes with enhanced triggered activity.

Impaired rhythmic spontaneous calcium release from the SR (calcium clock) plays a pivotal role in SAN dysfunction in HF [23]. HF suppresses the calcium clock, which is characterized by an attenuated intrinsic heart rate, late diastolic calcium elevation, and superior shift to isoproterenol and caffeine stimulation [97]. Furthermore, these findings suggest that AF is a disorder caused by a defective and inefficacious SAN. The SAN is really sick not only in sick sinus node syndrome but also in the vast majority of cases of AF occurring in the elderly. However, the SAN function is not usually explored in elderly patients with AF.

### 3.5. HF-enhanced Fibrosis and Stretch Differentially Regulates PV and SAN Electrical Activity

Fibrosis is common in HF SAN and PVs. However, fibrosis may have different effects on the SAN compared to that on PVs. In a rabbit model, our study revealed that the SAN in HF exhibited an automaticity exit block, SAN-PV conduction block, and severe fibrosis (Figure 4) [90]. Therefore, the abnormal conduction may be attributed to increased fibrosis in SAN [98]. Differently, extracellular protein matrix (collagen) may increase PV arrhythmogenesis through calcium overload via the activation of P38 [62].

The increase in left ventricular end diastolic pressure in HF causes an increase in the intra-atrial pressure and atrial stretch, which not only results in atrial electrical remodeling but also induces atrial dilatation [99,100]. Increased intra-atrial pressure caused by HF induces PV electric remodeling. Dilated PVs enhance PV arrhythmogenesis through the mechanoelectrical feedback [58,59]. Stretch of the SAN increases heart rate, however, dilation of RA is unusual in HF [101,102]. The higher expression level of heat shock protein-70 caused by the effects of low-oxygenated blood provides RA protective effects against oxidative stress and inflammation [103]. It is unclear why HF on the left side of the heart causes dysfunction in the SAN located on the right side of the heart [98]. Both neurohormonal activation and atrial stretch may play a critical role [93].

### 3.6. Electrolyte Disturbance Differentially Regulates PV and SAN Electrical Activity

Electrolyte disturbance is common in HF patients since hypokalemia and/or hyponatremia may be caused by the use of diuretics or fluid overload. Hypokalemia results in a reduction in the SAN and PV beating rates but induces PV burst firings and DADs. Hyponatremia induces PV burst firings and DADs but exhibits a minimal effect on the SAN and PV beating rates [104]. Accordingly, electrolyte imbalance differentially regulates SAN and PV electrical activity, which may contribute to the high risk of AF in HF.

## 4. Therapeutic Implication for AF

Patients with HF may have electrophysiological evidence of impaired SAN function even in the absence of clinical features of sick sinus syndrome [93]. This condition may be aggravated by the use of negative chronotropic drugs. β-blockers are a standard treatment for HF and the first-line choice for patients with HF and concomitant AF with a rapid ventricular response; however, β-blockers have not demonstrated a substantial reduction in mortality and HF hospitalization [7,105]. Digoxin reduces HF-related hospitalization and is recommended as the second-line choice for rate control of AF in HF [7]. However, in the presence of β-blocker and digoxin, clinical SAN dysfunction is common and may reflect an impaired SAN function in patients with HF [93]. Propafenone causes more severe SAN dysfunction because of its additional β-blocking effect [66]. Dronedarone is also contraindicated in patients with HF and SAN dysfunction. Therefore, antiarrhythmic drugs for rhythm control in patients with AF and HF are limited to amiodarone [105]. However, amiodarone exhibits inhibitory effects on multiple ion channels (I_to_, I_Kr_, I_Ks_, I_Ca-L_, and I_Na_) and the β-adrenergic system, which may impair SAN function. Consequently, in the presence of SAN dysfunction, the use of all antiarrhythmic drugs is often restricted because of the risk of deteriorating SAN dysfunction [66].

Ivabradine is licensed for the treatment of HF [106]. Ivabradine inhibits the I*_f_* in cardiac pacemaker cells and reduces the heart rate [107]. Ivabradine increases stroke volume by increasing the diastolic time, reduces myocardial oxygen demand, reverses LV remodeling, and prevents disease progression in patients with HF [108,109]. However, there is accumulating data indicating that there is increased risk of AF incidence during ivabradine treatment. The SHIFT, BEAUTIFUL, and SIGNIFY trials, three landmark trials of ivabradine, all revealed an increased incidence of AF in the group with ivabradine, compared with the placebo group [110,111,112]. In a meta-analysis of eight randomized, controlled trials involving patients with HF or chronic coronary artery disease, ivabradine was associated with a 15% increase in the relative risk of AF [113]. Moreover, an analysis revealed a 39% higher risk of AF in patients who had received an aggressive dosage regimen of ivabradine than in control patients [112]. Accordingly, the inhibition of I*_f_* by ivabradine may increase the risk of AF in patients with HF. The highlights of prescribing information suggest that ivabradine-treated patients should receive regular monitoring for the occurrence of AF. The normal SAN and HF SAN were reported to exhibit different responses to ivabradine. Ivabradine causes a greater reduction in the pacemaker rate and diastolic depolarization rate in the HF SAN than in the normal SAN, which suggests that the HF SAN can be more susceptible to the effects of ivabradine [90]. Moreover, ivabradine increases the incidence of SAN automaticity exit block and SAN-PV conduction block in HF (Figure 5) [90]. Accordingly, ivabradine reduces the SAN rate and disrupts the SAN-PV electrical connection in HF, leading to an increase in ectopic foci in the PVs by avoiding the overdriving suppression from the SAN and an increase in AF risk in HF.

Ranolazine is used to treat chronic angina. Ranolazine attenuates sodium-dependent calcium overload by inhibiting I_Na-Late_ and reduces tension in the heart wall, leading to reduced oxygen requirement during myocardial ischemia [114]. Ranolazine has been reported to exhibit antiarrhythmic effects in studies on ischemia and HF [115,116]. An animal study revealed that ranolazine inhibited the SAN function in HF and normal rabbits, suggesting that I_Na-Late_ can have a role in the electrophysiology of the SAN [96]. However, whether ranolazine might induce SAN dysfunction in clinical HF setting remains unanswered.

## 5. Conclusions

HF is commonly associated with AF and SAN dysfunction. HF induces chronic adrenergic stimulation, neurohormonal activation, abnormal intracellular calcium handling, elevated cardiac filling pressure with atrial stretch, and fibrosis, which cause electrical and structural remodeling of PVs and the SAN and contribute to AF and SAN dysfunction. PVs exhibit a higher level of arrhythmogenesis than LA in HF. HF-enhanced PV arrhythmogenesis might play a critical role in the initiation and maintenance of AF. The effects of HF on the SAN modulate the electro-pharmacological responses of the SAN, which may result in SAN dysfunction. In the presence of SAN dysfunction, the use of antiarrhythmic drugs for treating AF in HF patients has been restricted, and these drugs should be used cautiously.

## Figures and Tables

**Figure 1 ijms-20-03224-f001:**
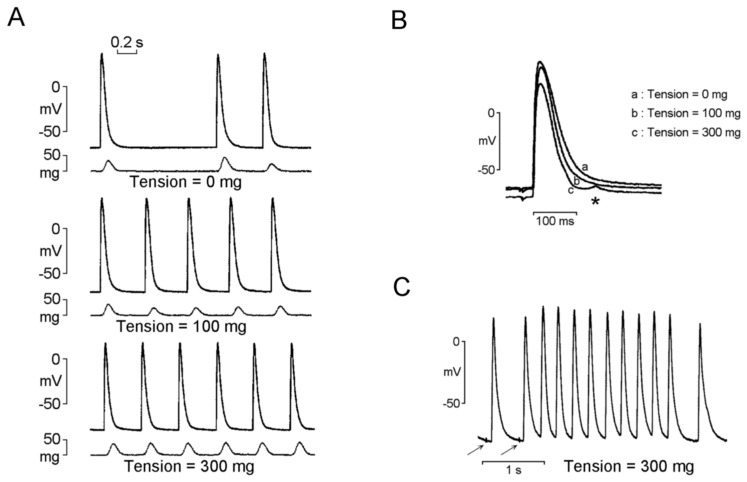
Effects of stretch on the electrical activity and the action potential (AP) configuration of the pulmonary veins (PVs). (**A**) Stretch force dependently increased the firing rate of the spontaneous activity of the PVs. (**B**) Superimposed tracings of PVs in which stretch force dependently decreased the amplitude and duration of the AP and induced delayed afterdepolarization (asterisk). (**C**) Stretch induced early afterdepolarization and burst firings in PVs. Arrow indicates electrical stimuli (2 Hz). “Modified with permission from Chang, S.L., et al. [58]”.

**Figure 2 ijms-20-03224-f002:**
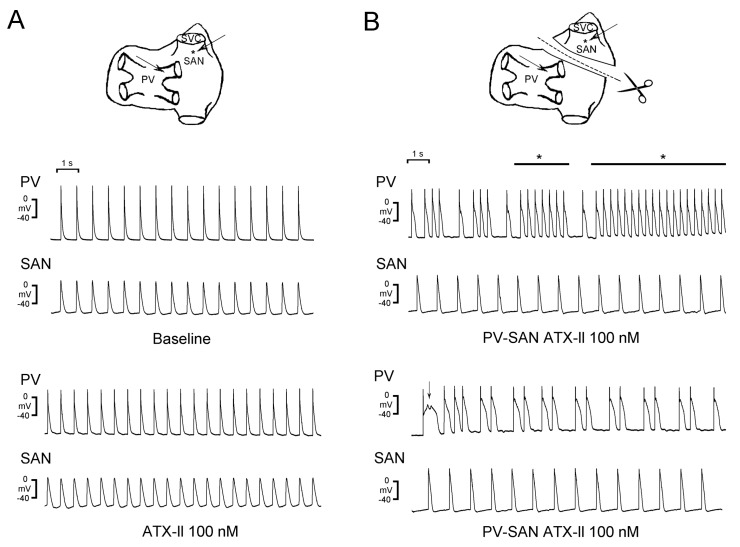
Sinoatrial node (SAN)-pulmonary vein (PV) electrical interaction. The schematic drawings show the simultaneous recordings (arrows) at the SAN and PVs in intact (Panel **A**) and disconnected (Panel **B**) SAN-PV preparations before and after the treatment of Anemonia sulcata toxin (ATX)-II. Burst firings (right middle panel) and early afterdepolarizations (EADs, right bottom panel) were induced in isolated PV preparation after being separated and superfused with ATX-II. The asterisks indicate burst firings and the arrowhead indicates EAD. “Modified with permission from Chen, Y.C., et al. [26]”.

**Figure 3 ijms-20-03224-f003:**
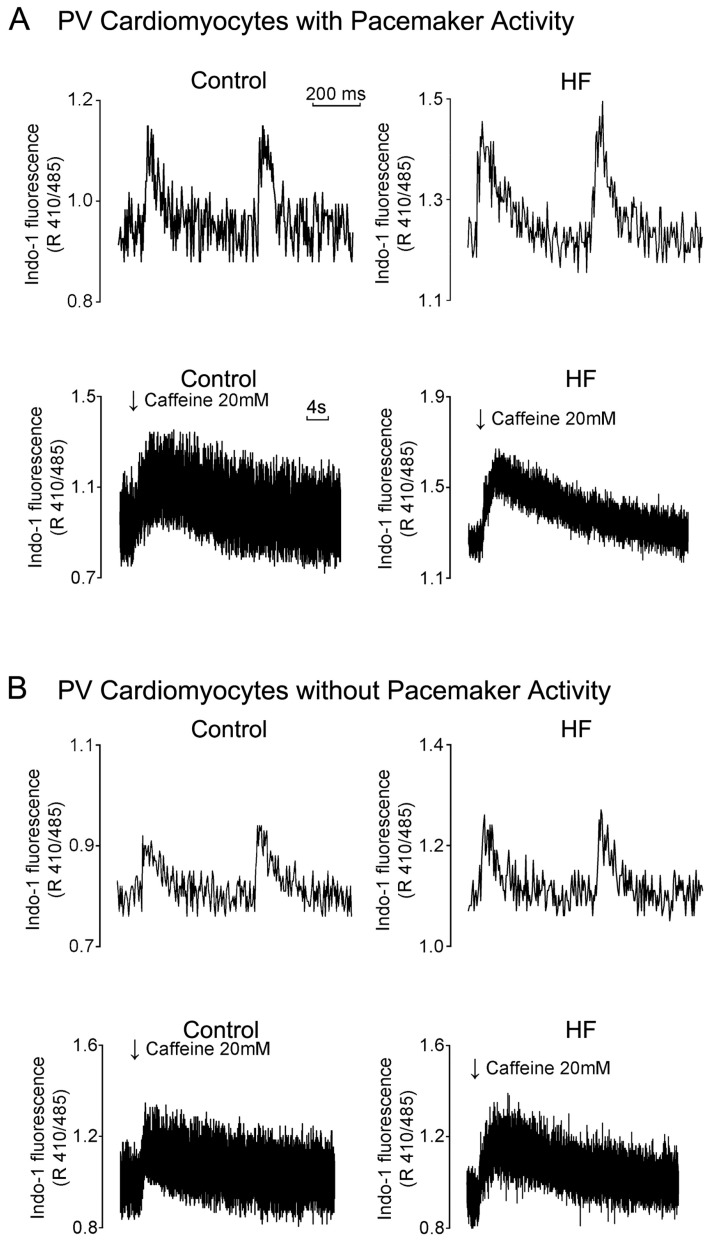
Intracellular Ca^2+^ transient (Ca^2+^i) and the calcium stores from the control and heart failure (HF) pulmonary vein (PV) cardiomyocytes. HF PV cardiomyocytes with (panel **A**) or without (panel **B**) pacemaker activity have a larger Ca^2+^i and calcium stores measured from caffeine (20 mM)-induced Ca^2+^i than control PV cardiomyocytes with or without pacemaker activity, respectively. “Modified with permission from Chang, S.L., et al. [91]”.

**Figure 4 ijms-20-03224-f004:**
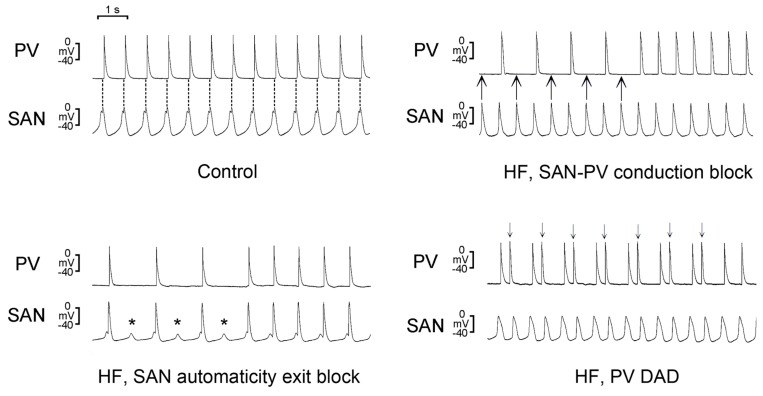
Effects of heart failure (HF) on sinoatrial node (SAN) electrical activity and pulmonary vein (PV) arrhythmogenesis. There is SAN to PV electric conduction in the control SAN-PV preparation (top left panel). SAN automaticity exit blocks (asterisks, bottom left panel) and SAN-PV conduction blocks (arrows, top right panel) with absence of PV electrical activity were found in HF SAN-PV preparations. Bottom right panel shows delayed afterdepolarizations (DADs; arrows) in a HF SAN-PV preparation. In top left panel, dashed lines indicate the peaks of SAN electrical activity. “Modified with permission from Chan, C.S., et al. [90]”.

**Figure 5 ijms-20-03224-f005:**
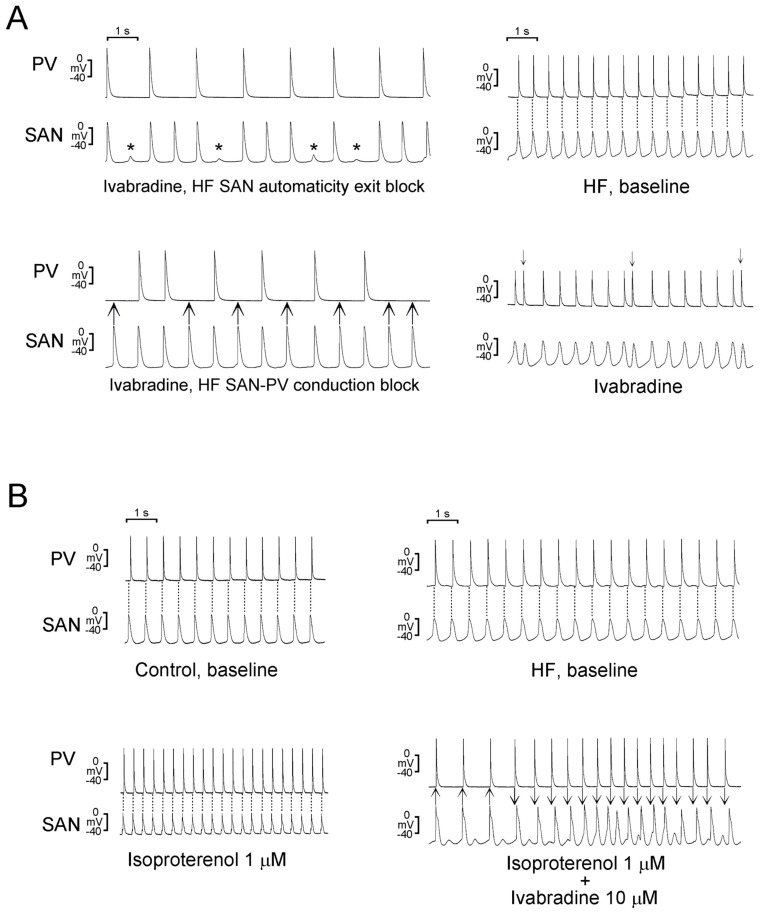
Effects of ivabradine on the electrical activity of control and heart failure (HF) sinoatrial node (SAN)-pulmonary vein (PV) preparations and PV arrhythmogenesis. (**A**) SAN automaticity exit blocks (asterisks) (top left panel) and SAN-PV conduction blocks (arrows) (bottom left panel) in HF SAN-PV preparations after ivabradine administration. Bottom right panel shows delayed afterdepolarizations (arrows) in HF SAN-PV preparations after ivabradine administration. (**B**) Left panel shows that isoproterenol (1 µM) accelerated electrical activity in SAN-PV preparation without change of the direction of SAN-PV electrical conduction. Right panel shows that the direction of electrical conduction between the SAN and PV reversed in HF SAN-PV preparation in the presence of isoproterenol (1 µM) with ivabradine (10 µM). Dashed lines indicate the peaks of SAN electrical activity. “Modified with permission from Chan, C.S., et al. [90]”

**Table 1 ijms-20-03224-t001:** The distinct electrophysiological characteristics of pulmonary veins and sinoatrial node.

	Calcium Regulation	Pacemaker Current	Connexin			Stretch Channel	Vascular Property	Autonomic Control
			40	43	45			
PVs	+++	+	+	++	++	+	+	+
SAN	++	++	+	-	++	+	-	+

PVs = pulmonary veins; SAN = sinoatrial node.

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
