# Peer review of "Heart Failure Differentially Modulates Natural (Sinoatrial Node) and Ectopic (Pulmonary Veins) Pacemakers: Mechanism and Therapeutic Implication for Atrial Fibrillation"

_ijms, 2019, doi:10.3390/ijms20133224_

Round 1

Reviewer 1 Report

The topic is interesting, and the manuscript is well structured. I have a few comments and suggestions.

1. Section 2. I think that there were few prior studies investigating structural characteristics of the PV sleeves in humans. I found the following papers recently published. Please discuss PV sleeves more.

Pulmonary Vein Sleeve Length and Association With Body Mass Index and Sex in Atrial Fibrillation. Ellis CR, Saavedra P, Kanagasundram A, Estrada JC, Montgomery J, Farrell M, Shen S, Crossley GH, Michaud G, Shoemaker MB. JACC Clin Electrophysiol. 2018 Mar;4(3):412-414.

Structural relation between the superior vena cava and pulmonary veins in patients with atrial fibrillation. Yoshida K, Baba M, Hasebe H, Shinoda Y, Harunari T, Ebine M, Uehara Y, Watabe H, Takeyasu N, Horigome H, Nogami A, Ieda M. Heart Vessels. 2019 May 22. doi: 10.1007/s00380-019-01431-z.

2. Ref. 52 Tsao et al. In this study, PV size could not predict AF firing. Only 28% patients showed arrhythmogenic foci from the largest PV. Please comment this discrepancy between the studies.

3. Section 3.1 This section is very important. Please describe more extensively. Electrical, structural, and autonomic remodeling should contribute to SAN dysfunction in patients with AF. I think this section should be the core of this review article from the point of view of clinical electrophysiology.

4. Section 3.2  I am very interesting in this section (crosstalk between PVs and SAN). However, there are only 2 citations presented here (No 66 and 67). Does this finding derive only from the authors’ studies? Are there more studies supporting this finding? What are mechanisms for this crosstalk? What is clinical implication of this finding?

5. line 252 typo. "a greater longer width"

Author Response

Responses to the reviewer 1

   Thank you very much for your detailed comments, and we appreciate your helpful comments and for allowing us to revise this manuscript. Those comments are very instructive and very helpful to this manuscript. The responses to those comments are dictated below and these messages have been added in the text:

1.     Regarding the specific comment “Section 2. I think that there were few prior studies investigating structural characteristics of the PV sleeves in humans. I found the following papers recently published. Please discuss PV sleeves more. Pulmonary Vein Sleeve Length and Association with Body Mass Index and Sex in Atrial Fibrillation. Ellis CR, Saavedra P, Kanagasundram A, Estrada JC, Montgomery J, Farrell M, Shen S, Crossley GH, Michaud G, Shoemaker MB. JACC Clin Electrophysiol. 2018 Mar;4(3):412-414. Structural relation between the superior vena cava and pulmonary veins in patients with atrial fibrillation. Yoshida K, Baba M, Hasebe H, Shinoda Y, Harunari T, Ebine M, Uehara Y, Watabe H, Takeyasu N, Horigome H, Nogami A, Ieda M. Heart Vessels. 2019 May 22. doi: 10.1007/s00380-019-01431-z.”

   We appreciate this comment very much. We agreed with your comment that we should discuss PV sleeves more. According to your suggestions, we have included the recent published papers (Ellis el al., JACC Clin Electrophysiol 2018, and Yoshida et al., Heart Vessel 2019) and made more discussions in the revised manuscript (page 2, line 67-72, red font) as follows: “Previous histological examination in autopsy specimens has shown that PV myocardial sleeves were found in 100% of patients with AF, compared to 85% of patients without AF [Hassink, et al., J Am Coll Cardiol. 2003]. The recent studies from voltage map showed that the lengths of myocardial sleeves are longer in the left and right superior PVs but markedly shorter in the left and right inferior PVs [Ellis, et al. Clin Electrophysiol. 2018; Yoshida, et al., Heart vessels. 2019]. Patients with AF have significant longer myocardial sleeves [Hassink, et al., J Am Coll Cardiol. 2003]. The length of PV myocardial sleeves correlates positively with male sex, body mass index and body surface area [Ellis, et al. JACC Clin Electrophysiol. 2018; Yoshida, et al., Heart vessels. 2019].

2.     Regarding the specific comment “Ref. 52 Tsao et al. In this study, PV size could not predict AF firing. Only 28% patients showed arrhythmogenic foci from the largest PV. Please comment this discrepancy between the studies.”

   We appreciate the comment very much. Significant dilation of both superior PVs was demonstrated in patients with AF, however, only 28% of the trigger foci arose from the largest PVs. Accordingly, PV size can’t predict AF firing [Tsao, et al., J Cardiovasc Electrophysiol. 2001]. In clinical series, longer PV myocardial sleeves corresponded to the relatively higher number of foci identified [Haïssaguerre et al, NEJM. 1998; Chen, et al., Circulation. 1999], this finding suggests a relationship between the extent of PV myocardial sleeves and the initiation source of ectopic foci. It was speculated that the anatomic and geometric differences of PVs may participate in the firing of PVs in AF [Tsao, et al., J Cardiovasc Electrophysiol. 2001]. According a to your suggestions, we have added additional discussions in the revised manuscript (page 4, line 148-153, red font) as follows Significant dilation of both superior PVs was demonstrated in patients with AF, however, only 28% of the trigger foci arose from the largest PVs. [Tsao, et al., J Cardiovasc Electrophysiol. 2001]. Previous studies have shown that higher number of AF foci arise from relatively longer PV myocardial sleeves, [Haïssaguerre et al, NEJM. 1998; Chen, et al., Circulation. 1999], suggesting a relationship between the extent of PV myocardial sleeves and ectopic foci of AF initiation. It was speculated that the anatomic and geometric differences of PVs may participate in the firing of PVs in AF, and PV size may not sufficiently predict the origin of AF firing [Tsao, et al., J Cardiovasc Electrophysiol. 2001].

3.     Regarding the specific comment “Section 3.1 This section is very important. Please describe more extensively. Electrical, structural, and autonomic remodeling should contribute to SAN dysfunction in patients with AF. I think this section should be the core of this review article from the point of view of clinical electrophysiology.”

   We appreciate the comment very much. We agreed with your comment that electrical, structural and autonomic remodeling should contribute to SAN dysfunction in patients with AF. In a canine model, atrial tachypacing has been shown to downregulate the mRNA expression of HCN4 and reduce SAN If [Yeh, et al., Circulation. 2009], suggesting that AF results in electrical remodeling of SAN. Besides, AF induced by rapid atrial pacing is associated with atrial structural change characterized by marked bi-atrial dilation, an increase in mitochondrial size and number, and disruption of the sarcoplasmic reticulum [Morillo, et al., Circulation. 1995]. The atrial dilation combined with rapid atrial rate may predispose to atrial ischemia and SAN dysfunction [Kezerashvili, et al., J Atr Fibrillation. 2008]. In addition, loss of muscle fibers in the SAN was found in AF patients in an autopsy study. [Davies, et al., Br Heart J. 1972]. The structural remodeling of SAN from repeated episodes of AF or prolonged persistence of AF can result in atrial cardiomyocyte apoptosis with progressive atrial fibrosis and dilation [John et, al., Circulation. 2016]. Accordingly, structural remodeling of SAN can contribute to SAN dysfunction in patients with AF.

   Autonomic nervous system is a major regulatory factor of SAN automaticity and sinoatrial conduction. Autonomic dysfunction is a common cause of SAN dysfunction [Kang, et al., Circulation 1981]. Adrenergic or cholinergic dysregulation may contribute to pacemaker and conduction abnormality within the SAN [Fedorov, et al., Circulation 2010]. According to your suggestions, we have provided those in the revised manuscript (page 5, line 202-215, red font).

4.     Regarding the specific comment “Section 3.2 I am very interesting in this section (crosstalk between PVs and SAN). However, there are only 2 citations presented here (No 66 and 67). Does this finding derive only from the authors’ studies? Are there more studies supporting this finding? What are mechanisms for this crosstalk? What is clinical implication of this finding?”

   We appreciate the comment very much. Slow heart rate predicts occurrence of AF [Grundvold, et al., Circ Arrhythm Electrophysiol. 2013; Hai, et al., Heart Lung Circ. 2015] and recurrence of AF after catheter ablation [Wu, et al., Am J Cardiol. 2018]. Clinical studies have shown that SAN dysfunction is frequently associated with the genesis of AF and atrial flutter, and “bradycardia-tachycardia syndrome” [Ferrer. JAMA. 1968; Gomes, Circulation. 1981]. Up to 50% of patients with SAN dysfunction are accompanied by AF [Hocini, et al., Circulation. 2103; Lamas, et al., N Engl J Med. 2002]. It is speculated that SAN dysfunction and AF share similar risk factors and pathophysiological process. In a guinea-pig model, the action potentials recorded from PVs were dominated by the SAN, and rapid pacing in PVs could overdrive the SAN [Cheung, J. Physiol. 1981]. This laboratory evidence suggested putative crosstalk between PVs and the SAN. According to your suggestions, we have provided those in the revised manuscript as follows (page 5, line 221-222; page 6, line 223-229, red font).

5.     Regarding the specific comment “line 252 typo. "a greater longer width"”

   We appreciate the comment very much. We have corrected the mistake in the revised manuscript (page 7, line 295, red font) as follows: “HF PV cardiomyocytes have a greater width, longer duration and longer decay time of the calcium transient (Figure 3).”

The above descriptions are the responses to your comments and suggestions.

Sincerely yours,

Yi-Jen Chen, MD, PhD

Graduate Institute of Clinical Medicine, Taipei Medical University

Reviewer 2 Report

Commentary

The article  is  very interesting, however   I would like to express some objections concerning some major  points addressed by the authors

Abstract

“calcium dysregulation”. Authors should eliminate this reference to serum calcium which is questionable. Troubles of serum   calcium   may be present in heart failure, but do not represent a qualifying element. Ion channel disorders are another thing, so please eliminate "calcium dysregulation"

which are ectopic pacemakers, Please delete. The statement is groundless. It could be replaced by "which are the point of onset of ectopic electrical activity"

Main document

calcium dysregulation: please  remove this assertion

Introduction, row 43  “The  coexistence of HF and AF increases the mortality and morbidity of AF and HF, respectively” Please remove this phrase. Indeed, this generic assertion is contradicted by the results of a recent registry study ,that would exclude that in  HFREF patients with AF  the  composite of  all-cause death and  HF hospitalizations  is worse compared to HFREF patients with sinus rhythm. The pertinent  reference is  reported below: Zafrir B , et al; ESC-HFA HF Long-Term Registry Investigators. Prognostic implications of atrial fibrillation in heart failure with reduced, mid-range, and preserved ejection fraction: a report from 14 964 patients in the European Society of Cardiology Heart Failure Long-Term Registry. Eur Heart J. 2018 Dec 21;39(48):4277-4284.

Introduction  “ AF has been identified as a crucial predictor of mortality in patients with HF. This is particularly   relevant to patients with less advanced HF and recent AF onset” The authors' statements do not take into account the fact that in patients with or without cardiac decompensation nowadays  in developed countries, it is possible to adopt strategies that limit the damage caused by AF. Thus an AF treated with rate control is not associated with more unfavorable clinical outcomes compared to a patient with sinus rhythm maintained and protected with the use of the rhythm control strategy. In the Introduction, please add some considerations in this regard.   .

EAD and DAD: Please clarify the meaning of these acronyms

SAN dysfunction in AF. Please  emphasize   this concept   adequately, by asserting that the  tachy-brady syndrome is the extreme expression of a continuum, characterized by a substantial loss of integrity of the SAN function. In other words, it can be asserted that atrial fibrillation is a disease also caused by a defective function of the SAN.   If SAN could  be less torpid , it would capable of antagonizing the tendentially predominating activity of pulmonary veins, and atrial fibrillation would not arise. Please address this controversial issue in the discussion.

 Row 217 In the CASTLE-AF trial, all-cause mortality, cardiovascular death, and   hospitalization for HF were significantly reduced by catheter ablation (PV isolation: 100%, additional  lesions: 51.7%), suggesting that PV trigger might play a crucial role in the initiation of AF in patients  with HF [72]. These sentences are questionable. For example, it has not been recognized yet the detrimental role that the massive destruction of atrial cardiomyocites by radiofrequency( RF)  can play, thus disturbing the mechanical activity of atrial chamber, that is the region that RF ablation would aim to protect . In this regard, please consult the dissonant voices of other authors: Packer M Eur Heart J. 2019 Jun 14;40(23):1873-1879.

. These findings suggest that HF differentially regulates SAN and PV calcium   homeostasis. The authors could be more courageous by asserting tha AF is a disorder caused by a defective and inefficacious SAN. The sinus node is really sick not only in sick sinus syndrome but also in the vast majority of cases of atrial fibrillation occurring in the elderly. This aspect does not emerge because the SAN function is not usually explored in elderly  patients with AF.

Row 316 Ivabradine is licensed for the treatment of HF. The authors would be awarded by my approval, if they would  remark that ivabradine is an arrhythmogenic agent,able to elicit AF in HF patients. Thus its use is questionable in CHF patients proven to be  prone to relapses of AF. Please emphasize this aspect, that contradicts the indication of ivabradine for CHF, instead  resulting from societal guidelines.

Author Response

Responses to the reviewer 2

   Thank you very much for your detailed comments, and we appreciate your helpful comments and for allowing us to revise this manuscript. Those comments are very instructive and very helpful to this manuscript. The responses to those comments are dictated below and these messages have been added in the text:

Abstract

1.     Regarding the specific comment “"calcium dysregulation”. Authors should eliminate this reference to serum calcium which is questionable. Troubles of serum calcium may be present in heart failure, but do not represent a qualifying element. Ion channel disorders are another thing, so please eliminate "calcium dysregulation"”

   We appreciate the comment very much. We agreed with your comment that “calcium dysregulation” is questionable. Troubles of serum calcium may be present in HF, but do not represent a qualifying element. Ion channel disorders are another thing. We have corrected the mistake in the revised manuscript as your suggestion (page 1, line 25) as follows: “HF is associated with chronic adrenergic stimulation, neurohormonal activation, abnormal intracellular calcium handling, elevated cardiac filling pressure with atrial stretch, and fibrosis.”

2.     Regarding the specific commentwhich are ectopic pacemakers, Please delete. The statement is groundless. It could be replaced by "which are the point of onset of ectopic electrical activity"”

   We appreciate the comment very much. We agreed with your comment. We have revised the manuscript (page 1, line 26-27, red font) as follows: “Pulmonary veins (PVs), which are the points of onset of ectopic electrical activity, are the most crucial AF triggers.”

Main document

1.     Regarding the specific comment “calcium dysregulation: please remove this assertion”

   We appreciate the comment very much. We agreed with your comment and have removed this assertion and replaced it with “abnormal intracellular calcium handling in the revised manuscript (page 1, line 43; page 3, line 3-124; page 13, line 400).

2.     Regarding the specific comment “Introduction, row 43 “The coexistence of HF and AF increases the mortality and morbidity of AF and HF, respectively” Please remove this phrase. Indeed, this generic assertion is contradicted by the results of a recent registry study, that would exclude that in HFREF patients with AF the composite of all-cause death and HF hospitalizations is worse compared to HFREF patients with sinus rhythm. The pertinent reference is reported below: Zafrir B, et al; ESC-HFA HF Long-Term Registry Investigators. Prognostic implications of atrial fibrillation in heart failure with reduced, mid-range, and preserved ejection fraction: a report from 14 964 patients in the European Society of Cardiology Heart Failure Long-Term Registry. Eur Heart J. 2018 Dec 21;39(48):4277-4284.”

   We appreciate the comment very much. We agreed with your comment and have removed this phrase in the revised manuscript as your suggestion. We also have provided your recommended reference in the revised manuscript and made discussions (page 2, line 45-48, red font) as follows:A recent registry study showed that AF increased the risk for the composite of mortality and HF hospitalization in HF patients with preserved or mid-range ejection fraction, but not in HF patients with reduced ejection fraction after multivariable adjustment [Zafrir, et al., Eur Heart J. 2018].

3.     Regarding the specific comment “Introduction “AF has been identified as a crucial predictor of mortality in patients with HF. This is particularly relevant to patients with less advanced HF and recent AF onset” The authors' statements do not take into account the fact that in patients with or without cardiac decompensation nowadays in developed countries, it is possible to adopt strategies that limit the damage caused by AF. Thus, an AF treated with rate control is not associated with more unfavorable clinical outcomes compared to a patient with sinus rhythm maintained and protected with the use of the rhythm control strategy. In the Introduction, please add some considerations in this regard.”

   We appreciate the comment very much. We agreed with your comment. Adopted strategies, such as standard heart failure therapy, stroke prevention, rate or rhythm control, limit the damage caused by AF [Kirchhof, et al., Europace. 2016]. Clinical trials have failed to demonstrate superiority of either a rate or rhythm-control strategy in patients with AF and concomitant HF [Roy, et al., N. Engl. J. Med. 2008]. An AF patient with HF treated with rate control is not associated with more unfavorable clinical outcome compared to a patient with sinus rhythm maintained and protected with the use of the rhythm control strategy. According to your suggestions, we have provided those in the revised manuscript (page 2, line 48-51, red font) as followsAn AF treated with rate control is not associated with more unfavorable clinical outcomes compared to a patient with sinus rhythm maintained and protected with the use of the rhythm control strategy [Roy, et al., N. Engl. J. Med. 2008]. It is possible to adopt strategies that limit the damage caused by AF such as standard heart failure therapy, and stroke prevention [Kirchhof, et al., Europace. 2016].

4.     Regarding the specific comment “EAD and DAD: Please clarify the meaning of these acronyms”

   We appreciate the comment very much. We have clarified the meaning of these acronyms in the revised manuscript (page 2, line 93-94; page 3, line 106, red font)

5.     Regarding the specific comment “SAN dysfunction in AF. Please emphasize this concept adequately, by asserting that the tachy-brady syndrome is the extreme expression of a continuum, characterized by a substantial loss of integrity of the SAN function. In other words, it can be asserted that atrial fibrillation is a disease also caused by a defective function of the SAN. If SAN could be less torpid, it would capable of antagonizing the tendentially predominating activity of pulmonary veins, and atrial fibrillation would not arise. Please address this controversial issue in the discussion.”

   We appreciate the comment very much. According to your suggestions, we have provided these discussions in the revised manuscript (page 5, line 215-219, red font) as follows: “Tachy-brady syndrome is the extreme expression of a continuum, characterized by a substantial loss of integrity of the SAN function. AF is a disease also caused by a defective function of the SAN. If SAN could be less torpid, it would be capable of antagonizing the tendentially predominating activity of PVs, and AF would not arise.

6.     Regarding the specific comment “Row 217 In the CASTLE-AF trial, all-cause mortality, cardiovascular death, and hospitalization for HF were significantly reduced by catheter ablation (PV isolation: 100%, additional lesions: 51.7%), suggesting that PV trigger might play a crucial role in the initiation of AF in patients with HF [72]. These sentences are questionable. For example, it has not been recognized yet the detrimental role that the massive destruction of atrial cardiomyocytes by radiofrequency (RF) can play, thus disturbing the mechanical activity of atrial chamber, that is the region that RF ablation would aim to protect. In this regard, please consult the dissonant voices of other authors: Packer M Eur Heart J. 2019 Jun 14;40(23):1873-1879.”

   We appreciate the comment very much. We agreed with your comment that massive destruction of atrial cardiomyocytes by radiofrequency (RF) can disturb the mechanical activity of atrial chamber. According to your suggestions, we have provided those in the revised manuscript (page 7, line 259-263, red font) as follows “However, the detrimental role that the massive destruction of atrial cardiomyocytes by radiofrequency can play, thus disturbing the mechanical activity of atrial chamber, that is the region that radiofrequency ablation would aim to protect. Catheter ablation can cause further injury to the LA and impair the reservoir, conduit and transport functions of the LA. The benefit-to-risk of catheter ablation in patients with HF remains to be established [Parker. Eur Heart J. 2019]..

7.     Regarding the specific comment “These findings suggest that HF differentially regulates SAN and PV calcium homeostasis. The authors could be more courageous by asserting that AF is a disorder caused by a defective and inefficacious SAN. The sinus node is really sick not only in sick sinus syndrome but also in the vast majority of cases of atrial fibrillation occurring in the elderly. This aspect does not emerge because the SAN function is not usually explored in elderly patients with AF.”

   We appreciate the comment very much. We agreed with your comment that AF is a disorder caused by a defective and inefficacious SAN. The SAN is really sick not only in sick sinus node syndrome but also in the vast majority of cases of AF occurring in the elderly. According to your suggestions, we have provided those in the revised manuscript (page 9, line 309-312, red font) as follows: “Besides, these findings suggest that AF is a disorder caused by a defective and inefficacious SAN. The SAN is really sick not only in sick sinus node syndrome but also in the vast majority of cases of AF occurring in the elderly. However, the SAN function is not usually explored in elderly patients with AF.”

8.     Regarding the specific comment “Row 316 Ivabradine is licensed for the treatment of HF. The authors would be awarded by my approval, if they would remark that ivabradine is an arrhythmogenic agent, able to elicit AF in HF patients. Thus its use is questionable in CHF patients proven to be prone to relapses of AF. Please emphasize this aspect, that contradicts the indication of ivabradine for CHF, instead resulting from societal guidelines.”

   We appreciate the comment very much. Ivabradine increases stroke volume by increasing the diastolic time, reduces myocardial oxygen demand, reverses LV remodeling and prevents disease progression in patients with HF [Reil, et al., J Am Coll Cardiol. 2013; Tardif, et al., Eur Heart J. 2011]. However, there is accumulating data indicating that there is increased risk of AF incidence during ivabradine treatment. In the highlights of prescribing information to use ivabradine, ivabradine is reported to increases the risk of AF. The SHIFT, BEAUTIFUL and SIGNIFY trials, three landmark trials of ivabradine, all revealed an increased incidence of AF in the group with ivabradine, compared with the placebo group [Swedberg, et al, Lancet 2010; Fox, et al, Eur Heart J. 2013; Fox, et al, Eur Heart J. 2015]. In a meta-analysis of eight randomized, controlled trials involving patients with HF or chronic coronary artery disease, ivabradine was associated with a 15% increase in the relative risk of AF [Martin, et al., Heart. 2104]. Moreover, an analysis revealed a 39% higher risk of AF in patients who had received an aggressive dosage regimen of ivabradine than in control patients [Fox, et al, Eur Heart J. 2015]. Accordingly, the inhibition of If by ivabradine may increase the risk of AF in patients with HF. According to your suggestions, we have emphasized this aspect in the revised manuscript (page 10, line 361-371; line 378-379, red font) as follows: “Ivabradine increases stroke volume by increasing the diastolic time, reduces myocardial oxygen demand, reverses LV remodeling and prevents disease progression in patients with HF [Reil, et al., J Am Coll Cardiol. 2013; Tardif, et al., Eur Heart J. 2011]. However, there is accumulating data indicating that there is increased risk of AF incidence during ivabradine treatment. The SHIFT, BEAUTIFUL and SIGNIFY trials, three landmark trials of ivabradine, all revealed an increased incidence of AF in the group with ivabradine, compared with the placebo group [Swedberg, et al, Lancet 2010; Fox, et al, Eur Heart J. 2013; Fox, et al, Eur Heart J. 2015]. In a meta-analysis of eight randomized, controlled trials involving patients with HF or chronic coronary artery disease, ivabradine was associated with a 15% increase in the relative risk of AF [Martin, et al., Heart 2104]. Moreover, an analysis revealed a 39% higher risk of AF in patients who had received an aggressive dosage regimen of ivabradine than in control patients [Fox, et al, Eur Heart J. 2015]. Accordingly, the inhibition of If by ivabradine may increase the risk of AF in patients with HF. The highlights of prescribing information suggest that ivabradine-treated patients should receive regular monitoring for the occurrence of AF.

The above descriptions are the responses to your comments and suggestions.

Sincerely yours,

Yi-Jen Chen, MD, PhD

Graduate Institute of Clinical Medicine, Taipei Medical University